# Preterm Infants on Early Solid Foods and Neurodevelopmental Outcome—A Secondary Outcome Analysis of a Randomized Controlled Trial

**DOI:** 10.3390/nu16101528

**Published:** 2024-05-19

**Authors:** Margarita Thanhaeuser, Fabian Eibensteiner, Melanie Gsoellpointner, Sophia Brandstetter, Renate Fuiko, Bernd Jilma, Angelika Berger, Nadja Haiden

**Affiliations:** 1Department of Pediatrics and Adolescent Medicine, Comprehensive Center for Pediatrics, Medical University of Vienna, 1090 Vienna, Austria; margarita.thanhaeuser@meduniwien.ac.at (M.T.); fabian.eibensteiner@meduniwien.ac.at (F.E.); sophia.brandstetter@meduniwien.ac.at (S.B.); renate.fuiko@meduniwien.ac.at (R.F.); angelika.berger@meduniwien.ac.at (A.B.); 2Department of Clinical Pharmacology, Medical University of Vienna, 1090 Vienna, Austria; melanie.gsoellpointner@meduniwien.ac.at (M.G.); bernd.jilma@meduniwien.ac.at (B.J.); 3Department of Neonatology, Kepler University Hospital, 4020 Linz, Austria

**Keywords:** preterm infants, solid foods, neurodevelopmental outcome

## Abstract

There are no evidence-based recommendations regarding the introduction of solid foods in preterm infants. The objective of this study was to investigate whether age at the introduction of solid foods affects neurodevelopmental outcomes. This study focuses on analyzing secondary outcomes from a prospective trial involving very low birth weight infants who were randomly assigned to either an early (10–12th week corrected age) or a late (16–18th week corrected age) complementary feeding group. The study evaluated neurodevelopmental outcomes at one and two years of corrected age, as well as at three years and four months of uncorrected age by utilizing Bayley scales. In total, 89 infants were assigned to the early and 88 infants to the late group, all with a mean gestational age of 27 + 1 weeks. A linear mixed-effects model was used to compare neurodevelopmental outcomes across the study groups, taking into account variables such as gestational age at birth, sex, nutrition at discharge, parents’ highest education level, and high-grade intraventricular hemorrhage. The analysis did not reveal any significant differences between the groups. The timepoint of the introduction of solid foods had no impact on neurodevelopmental outcomes at one and two years of corrected age, and at three years and four months of uncorrected age.

## 1. Introduction

There is still a lack of evidence-based guidelines on the introduction of solid foods in preterm infants, mainly due to the paucity of available randomized controlled trials [1,2]. Over two decades ago, Marriott et al. provided significant insights by comparing the early versus late introduction of complementary foods in preterm infants [3]. However, advances in infant care practices since then render these findings challenging to align with contemporary standards. Another randomized controlled trial was conducted in India in infants with a mean gestational age of 32 weeks providing only limited evidence applicable to extremely preterm infants in Western settings [4].

To address this gap in knowledge and to contribute to the development of evidence-based recommendations on the optimal timing of the introduction of solid foods in preterm infants, we conducted a randomized controlled trial of the early versus late introduction of standardized complementary foods in infants with a mean birth weight of <1000 g [5]. The primary outcome of height at one year corrected age did not differ between the study groups [5], and no notable differences were found in secondary outcomes and safety parameters such as iron and vitamin D status during the infants’ first year of life [6,7].

Brain growth and neurodevelopmental outcomes in preterm infants are significantly influenced by the nutrient composition of parenteral and enteral nutrition [8,9]. Early macro- and micronutrient deficiencies after birth can affect the developing brain in a number of ways, potentially affecting myelination, neurogenesis, neuronal growth, synaptogenesis, and basic neuronal metabolism [8,9]. While there are numerous publications examining early feeding interventions in both preterm and term infants and the subsequent outcomes, there is a paucity of the literature on interventions during the complementary feeding period and their influence on neurodevelopment, applicable to both term and preterm infants [10,11]. In a position paper on complementary feeding, the European Society for Pediatric Gastroenterology, Hepatology, and Nutrition (ESPGHAN) acknowledges insufficient data to form specific recommendations on the composition of solid foods especially for preterm infants, despite discussions on the importance of the adequate supply of iron-rich or -fortified foods and long-chain polyunsaturated fatty acids (PUFAs) for later neurodevelopment [12]. To provide more comprehensive guidance for promoting optimal neurodevelopment in infants, further research in this area is needed.

A predetermined secondary outcome of our randomized controlled trial was to explore whether the timing of introducing complementary feeding—either early or late—affects neurodevelopmental outcomes in infants during the initial three years of life.

## 2. Materials and Methods

This study focuses on analyzing secondary outcomes from a prospective, randomized, two-arm intervention trial of preterm infants on early solid feeding performed at a level IV neonatal care unit at the Medical University of Vienna, Austria. The study design and primary outcome as well as other secondary outcomes were recently published [3,4,5,10].

To sum up, the infants with a birth weight < 1500 g and a gestational age < 32 weeks were included starting from October 2013 to February 2020 and randomly assigned to either an early (10–12th week corrected age) or a late (16–18th week corrected age) feeding group at term equivalent age after informed consent was obtained from the parents. Because of the low risks to the participants, written informed consent from one parent was sufficient. The trial is registered on clinicaltrials.gov (NCT01809548) and was approved by the ethics committee of the Medical University of Vienna (EK: 1744/2012, date of approval 10 January 2013).

The infants with conditions that affect stable growth, i.e., gastrointestinal diseases such as necrotizing enterocolitis resulting in short bowel syndrome [13], Hirschsprung disease [14], and chronic inflammatory bowel disease [15], as well as those with bronchopulmonary disease [16], congenital heart disease [17], or major congenital birth defects or chromosomal aberrations were not eligible.

Infants were fed age-appropriate standardized complementary foods in addition to breastfeeding or formula feeding until the age of one year corrected for prematurity. Details on the standardized feeding boxes were described previously [5]. Five different food boxes following an age-based step-up concept containing commercially available ready-to-use baby jar food were available.

The primary aim of this secondary outcome analysis was the neurodevelopmental outcome of the infants at three years and four months of uncorrected age. Secondary outcomes included the assessment of neurodevelopment at one and two years of corrected age.

### 2.1. Study Visits and Assessment of Neurodevelopment

The families of the participating infants were invited to study visits together with the regular visits at the neonatal outpatient clinic at the expected due date, 6 weeks, 12 weeks, 6 months, 12 months, and 24 months of corrected age as well as at 3 years and 4 months of uncorrected age. Anthropometric measurements were collected at every visit.

Neurodevelopmental outcomes were assessed at one and two years of corrected age, and at three years and four months of uncorrected age using the Bayley scales of Infant-Toddler Development, third edition, German version [18,19]. The Bayley-III consists of five subtests: cognition, receptive and expressive communication, and fine and gross motor skills, and is used to measure the neurodevelopment of infants aged 16 days to 42 months. For each subtest, scaling values are calculated which range from 1 to 19 with a mean of 10 and a standard deviation of 3, based on normative data for the toddlers’ ages. The scores are converted into composite cognitive, language, and motor scores. These have a mean of 100 and a standard deviation (SD) of 15. Composite scores of −1 SD (values between 70 and 85) are defined as mild disability, and scores with −2 SD (values < 70) as severe disability. The tests were conducted and scored by two certified clinical psychologists with extensive experience in test administration.

### 2.2. Baseline Characteristics

Maternal and infant baseline characteristics as well as data on neonatal morbidity were collected from medical charts. Data on parental education were collected at the follow-up visits and divided into three groups according to the highest level of education of the child’s father or mother (primary, secondary, or tertiary education).

### 2.3. Statistical Analysis

In general, absolute and relative frequencies were calculated for ordinal and nominal data, respectively. For continuous variables, means and standard deviations or medians and interquartile ranges (IQRs) were calculated.

Statistical analysis was conducted using the R software (R Core Team 2022, www.R-project.org, accessed on 6 December 2022). Descriptive analysis was conducted using the absolute and relative frequencies for the ordinal or nominal data and means with standard deviations or medians with interquartile ranges (IQRs) for continuous data. A linear mixed-effects model was used to compare neurodevelopmental outcomes in terms of Bayley-III composite scores across the study groups, taking into account the following variables: study group, gestational age at birth, sex, nutrition at discharge, highest education of parents, and high-grade intraventricular hemorrhage (IVH, defined as >grade II) with a random intercept to adjust for possible correlation between the siblings of multiple births. The study group (i.e., early vs. late complementary feeding) was included to identify differences in neurological outcomes between the randomized studied complementary feeding groups. Gestational age at birth, sex, highest education of parents, and high-grade IVH were added as covariates to adjust for potential confounding of these factors on the studied patients’ neurological outcomes. All of these adjustment factors are on their own the known potential influencers of neurological development. This is also true for formula vs. mother’s own milk diet wherefore the covariate “nutrition at discharge” was added to the model as another potential empirical confounder. Model results are reported with estimated marginal means, 95% confidence intervals (95%CI), and *p*-values. Differences between neurodevelopmental outcome groups (no disability, mild disability, and severe disability) were calculated using chi-squared tests. *p* values < 0.05 were considered statistically significant.

## 3. Results

### 3.1. Screening and Participants

In total, 177 infants were randomized, 89 to the early group and 88 infants to the late group. After accounting for 21 dropouts for the per protocol analysis, 81 infants remained in the early group and 75 in the late group. At one year of corrected age, data from 152 (97.4%) infants were available (early group: *n* = 78, late group: *n* = 74). At two years of corrected age, data from 144 (92.3%) infants (early group: *n* = 72, late group: *n* = 72) and at three years four months data from 116 (74.5%) infants were available (early group: *n* = 52, late group: *n* = 64), respectively.

### 3.2. Baseline Characteristics and Neonatal Morbidity

Table 1 shows the maternal and infant baseline characteristics as well as data on neonatal morbidity. The study groups were similar regarding baseline characteristics and neonatal morbidity.

### 3.3. Primary Outcome

Neurodevelopmental outcome scores at three years and four months are shown in Table 2. The composite scores of all three categories were comparable between the study groups. Table 2 further displays values stratified according to the standard deviation of the composite scores in no, mild, and severe disability. Again, no differences between the groups could be found. Appendix A shows the model coefficients as well as the number of observations per model.

### 3.4. Secondary Outcomes

The analysis did not reveal any significant differences between the groups at one and two years of corrected age, which is depicted in Table 2. Post hoc, differences in neurodevelopmental outcome according to the type of feeding at discharge (Appendix A) and sex (Appendix A) were calculated. The type of nutrition at discharge had no significant influence on the neurodevelopmental outcome of the infants. At two years of corrected age, the male infants in the late group showed significantly lower scores in the cognitive (female: median 90; male: median 75) and language (female: median 84; male: median 66) assessments. However, this effect was no longer seen at 3 years and 4 months.

### 3.5. Lost to Follow-Up

Differences in baseline characteristics and neonatal morbidities between the infants with and without follow-up are shown in Appendix A. Infants without follow-up had a significantly higher gestational age at birth and a significantly higher birth weight.

## 4. Discussion

Age at the introduction of solid foods in preterm infants with a birth weight < 1500 g had no impact on neurodevelopmental outcome assessed with the Bayley-III scales at one and two years of corrected age, and at three years and four months of uncorrected age. No significant differences in the cognitive, language, and motor scores at any of the three timepoints could be found. Additionally, no discernible advantages of breastfeeding on the neurodevelopmental outcomes of the infants were identified in our cohort, contrary to the observations often described in the literature [20].

Therefore, the early introduction of solid foods in extremely preterm infants, a practice frequently adopted by families, can be considered safe. This safety extends not only to aspects such as growth, iron levels, and vitamin D status within the initial year of life but also encompasses later neurodevelopmental outcomes [5,6,7].

The results of this secondary outcome analysis align with expectations given the absence of persistent differences in macro- and micronutrient intake between the study groups, factors that might have otherwise accounted for an impact on neurodevelopmental outcomes [21].

This is one of the first randomized controlled studies to report on the neurodevelopmental outcome of VLBW preterm infants with a nutritional intervention in the complementary feeding period. Currently, only one of the two randomized controlled trials on complementary feeding in preterm infants available assessed the neurodevelopmental outcomes of infants. Gupta et al. investigated the introduction of solids at 4 versus 6 months of age in infants born before 34 weeks of gestational age [4]. At 12 months of corrected age, an Indian adaptation of the Bayley-II scales was performed, but no differences in neurodevelopmental outcomes between the study groups could be detected [4].

Data on term-born infants are similarly scarce, with most research focusing on the length of exclusive breastfeeding and its potential effects on neurodevelopmental outcomes later in life, rather than the specifics of introducing solid foods or their nutritional composition [22,23]. Jonsdottir et al. randomized term infants to an exclusive breastfeeding group until 4 months or 6 months of age. The objective of the study was to assess whether the duration of exclusive breastfeeding affected the neurodevelopmental outcomes of infants at 18 and 30–35 months of age, utilizing a parent-reported developmental status questionnaire and the Brigance Screens-II, an early childhood developmental assessment tool. The analysis revealed no significant differences between the study groups [22]. The PROBIT trial evaluated the neurodevelopmental outcome of term-born infants at 6.5 years of age with the strengths and difficulties questionnaire assessed by the parents and teachers of the infants and the Wechsler Abbreviated Scales of Intelligence. Despite variations in the duration of exclusive breastfeeding (either until 3 or 6 months of age), the trial concluded that the duration of breastfeeding did not impact later neurodevelopmental outcomes [23].

The collective findings from these studies suggest that the duration of exclusive breastfeeding, and consequently the age at which solid foods are introduced, does not exert any discernible influence on neurodevelopmental outcomes.

Given the small quantities of solid foods introduced during the initiation of complementary feeding, differences in nutrient intakes attributable to varying starting times appear too minimal to influence the outcome of our study. The window of opportunity for an optimization of brain development in infants due to nutritional interventions seems to be already closed by the time solid foods are introduced. Most data on feeding interventions are available from the early postnatal days, and less studies report on interventions in macro- and micronutrient intakes after discharge. Further research is needed to assess nutrient intakes during the complementary feeding period and its effect on growth and later development.

Nevertheless, an intriguing association emerges between exclusive breastfeeding at discharge and enhanced cognitive outcomes, even in cases of suboptimal initial weight gain, which is referred to as the “apparent breastfeeding paradox” [20]. Although breastfeeding may be linked to a significantly increased risk of losing weight z-score during hospitalization and head circumference z-scores at 2 years [24] and 5 years [20] of age, breastfeeding is also accompanied by a significantly reduced risk for suboptimal neurodevelopmental outcomes [20]. In our cohort, we did not see any advantages of breastfeeding in the neurodevelopmental outcome of infants, as Bayley-III composite scores were comparable between breast- and formula-fed infants at one and two years of corrected age and at 3 years and 4 months of uncorrected age.

Gender gaps favoring females in pre-school age are commonly reported [25,26]. At two years of corrected age, we observed significantly higher cognitive and language Bayley-III scores in the female infants of the late group. These differences did not persist until 3 years and 4 months of uncorrected age.

However, both analyses—differences in type of milk feedings and gender—were performed post hoc, and the study was not powered to detect any such differences. Thus, findings need to be interpreted cautiously.

Furthermore, the intrauterine heterogeneity of neurodevelopment needs to be discussed. Recent studies have focused on the differences in the developmental outcomes of preterm infants dependent on the phenotype of preterm birth [27,28].

Neurodevelopment appears to be altered already in utero due to the intrauterine environment and is therefore more than just a mere consequence of preterm birth. It is becoming increasingly evident that subtle changes in the intrauterine environment can have long-lasting effects on fetal programming and brain development. These changes may be influenced by a number of factors, including maternal nutrition, stress, inflammation, and medications [28,29,30]. Recent advancements in MRI imaging have unveiled that the functional connectivity of the fetal brain undergoes modifications prior to preterm birth, with a reduction in fetal brain functional connections observed in infants at a heightened risk of premature delivery [31,32].

However, the influence of prenatal factors on the neurodevelopmental outcomes of preterm infants has not been factored into our analyses. This consideration is pivotal for a comprehensive understanding of the complexity surrounding the neurodevelopmental trajectory of preterm neonates.

### 4.1. Improvement of Neurodevelopmental Outcome

At three years and four months of corrected age, the percentage of infants without any disability was rather low with 40–50% in the motor and language outcomes due to a high percentage of multilingualism and fine motor deficits and 70% in the cognitive outcome. About 20% of the infants showed a severe disability in cognitive, language, and motor development, which is comparable to the literature considering the low mean gestational age and birth weight of the infants included in our study.

It must be considered, however, that German norms were used for neurodevelopmental assessments, which tend to yield less favorable results compared to the American norms [33]. Additionally, while there was no selection bias in patient inclusion due to the randomized control design of the study, there was a follow-up bias. More patients with fewer comorbidities and higher birth weights and gestational ages were lost to follow-up, which might have influenced the outcome data.

Still, a meta-analysis by Pascal et al. from 2018 showed comparable rates of developmental delay. The authors included studies published over the past decade on infants with very low birth weight and reported a pooled prevalence of motor developmental delay of 20.6% and cognitive developmental delay of 16.9% [34].

The ongoing pursuit of enhancing neurodevelopmental outcomes in preterm infants remains a critical focus. The pivotal question centers around identifying interventions during the complementary feeding period that can be implemented to guarantee an optimal nutrient composition for the developing brain.

Identifying the optimal window for an optimization of nutrient intake to enhance the brain development and later neurodevelopmental outcomes of preterm infants is challenging. Age at the introduction of solid foods had no influence on outcome in our patients, nor on the growth, iron, and vitamin D status of the infants as recently published [5,6,7]. We suggest that the time of starting complementary feeding in preterm infants can be chosen according to the neurological abilities of the infant. Concerning neurodevelopmental outcomes of infants, the focus should be on the nutrient composition of solid foods and especially an adequate intake of iron and PUFAs rather than on age at weaning.

### 4.2. Limitations and Strengths

This paper is subject to several limitations, given that it is a secondary analysis of a randomized controlled trial and that the study was not designed to identify differences in neurodevelopmental outcomes. Furthermore, the characteristics of the patients lost to follow-up differed between the study groups. Infants without follow-up had a significantly higher birth weight and gestational age and were discharged ten days earlier due to fewer comorbidities. On the other hand, the breastfeeding rate at discharge was lower in infants who did not receive follow-up, and their parents tended to have lower education levels, factors that might have influenced neurodevelopmental outcomes.

Still, at three years and four months of uncorrected age, data from nearly 75% of the infants was available for analysis, which is a rather high follow-up rate. Another strength of the study is the unique patient collective of the infants with a mean birth weight < 1000 g. This is the first study on the neurodevelopmental outcome of extremely preterm infants after a nutritional intervention in the complementary feeding period.

## 5. Conclusions

The timing of the introduction of solid foods did not influence neurodevelopmental outcomes at one and two years of corrected age, and at three years and four months of uncorrected age. The early introduction of solid foods in extremely preterm infants can be regarded as safe, not only concerning growth, iron, and vitamin Dstatus within the first year of life, but also concerning later neurodevelopmental outcomes. Thus, decisions on the start of solid foods can be made according to the infants’ neurological abilities.

## Figures and Tables

**Table 1 nutrients-16-01528-t001:** Baseline characteristics and neonatal morbidity.

Parameter	Early Group (*n* = 81)	Late Group (*n* = 75)
*Obstetric and parental parameters*		
Multiple pregnancies	30 (37)	25 (33.3)
Cesarean delivery	70 (86.4)	73 (97.3)
Prenatal steroids (full course)	42 (51.9)	49 (65.3)
Premature rupture of membranes	34 (42)	32 (42.7)
Preeclampsia	8 (9.9)	7 (9.3)
Age of mother at birth	33 [±5]	33 [±7]
Age of father at birth	36 [±7]	36 [±7]
Education mother		
No graduation/school diploma	9 (11.1)	12 (16)
Middle school	29 (35.8)	22 (29.3)
Secondary school	11 (13.6)	17 (22.7)
Post-secondary school	29 (35.8)	22 (29.3)
Education father		
No graduation/school diploma	7 (8.6)	7 (9.3)
Middle school	36 (44.4)	31 (41.3)
Secondary school	10 (12.3)	10 (13.3)
Post-secondary school	25 (30.9)	21 (28)
*Neonatal parameters*		
Male sex	49 (60.5)	39 (52)
Gestational age (days)	190 [±16]–27 + 1	191 [±14]–27 + 2
Birth weight (g)	944 [±251]	938 [±259]
Small for gestational age	7 (8.6)	5 (6.7)
Gestational age (days) at discharge	261 [±18]–37 + 2	263 [±16]–37 + 4
Breast milk feeding at discharge	24 (29.6)	25 (33.3)
*Neonatal morbidity*		
NEC grade I and II	4 (4.9)	0 (0)
PDA	28 (34.6)	27 (36)
ROP ≥ grade III	5 (6.2)	3 (4)
IVH grade I and II	9 (11.1)	3 (4)
IVH ≥ grade III	3 (3.7)	5 (6.7)
PVL	0 (0)	2 (2.7)

Categorical data are presented as numbers with percentages in round parentheses. Continuous data are presented as the mean ± standard deviation in squared parentheses. IVH—intraventricular hemorrhage, NEC—necrotizing enterocolitis, PDA—persisting ductus arteriosus, PVL—periventricular leukomalacia, ROP—retinopathy of prematurity.

**Table 2 nutrients-16-01528-t002:** Bayley-III scales of infant development at 1 and 2 years of corrected and 3 years and 4 months of uncorrected age.

Parameter			Early Group	Late Group	*p*-Value
** *1 year of corrected age* **	**Cognitive**		*n* = 73	*n* = 73	
		Composite score	90 (80–105)	90 (80–105)	0.81
		No disability	55 (75.3)	49 (67.1)	0.706
		Mild disability	10 (13.7)	13 (17.8)	
		Severe disability	8 (11)	11 (15.1)	
	**Language**		*n* = 73	*n* = 73	
		Composite score	97 (83–103)	97 (84–103)	0.54
		No disability	52 (71.2)	50 (68.5)	0.996
		Mild disability	12 (16.4)	15 (20.5)	
		Severe disability	9 (12.3)	8 (11)	
	**Motor**		*n* = 74	*n* = 73	
		Composite score	100 (85–103)	92 (85–106)	1
		No disability	56 (75.7)	55 (75.3)	0.066
		Mild disability	13 (17.6)	9 (12.3)	
		Severe disability	5 (6.8)	9 (12.3)	
** *2 years of corrected age* **	**Cognitive**		*n* = 72	*n* = 71	
		Composite score	85 (75–100)	85 (70–105)	0.88
		No disability	39 (54.1)	38 (53.5)	0.689
		Mild disability	20 (27.8)	18 (25.3)	
		Severe disability	13 (18.1)	15 (21.1)	
	**Language**		*n* = 66	*n* = 66	
		Composite score	78 (56–94)	78 (62–96)	0.61
		No disability	26 (39.4)	28 (42.4)	0.146
		Mild disability	17 (25.8)	11 (16.7)	
		Severe disability	23 (34.8)	27 (40.9)	
	**Motor**		*n* = 69	*n* = 70	
		Composite score	89 (82–100)	89 (76–103)	0.67
		No disability	47 (68.1)	46 (65.7)	0.199
		Mild disability	18 (26)	13 (18.6)	
		Severe disability	4 (5.8)	11 (15.7)	
** *3 years 4 months of uncorrected age* **	**Cognitive**		*n* = 50	*n* = 61	
		Composite score	90 (85–100)	95 (80–100)	0.48
		No disability	35 (70)	41 (67.2)	0.650
		Mild disability	9 (18)	9 (14.8)	
		Severe disability	6 (12.5)	11 (18)	
	**Language**		*n* = 47	*n* = 59	
		Composite score	84 (72–94)	87 (72–94)	0.20
		No disability	20 (42.6)	30 (50.8)	0.584
		Mild disability	17 (36.2)	16 (27.1)	
		Severe disability	10 (21.3)	13 (22)	
	**Motor**		*n* = 48	*n* = 59	
		Composite score	82 (70–92)	84 (70–89)	0.98
		No disability	19 (39.6)	26 (44.1)	0.893
		Mild disability	17 (35.4)	19 (32.2)	
		Severe disability	12 (25)	14 (23.7)	

Composite scores are presented as median with the 25. and the 75. percentile in parenthesis. Data on disability are presented as the number of patients and percentage in parentheses. No disability is defined as Bayley-III composite scores > 85, mild disability as values 70–85, and severe disability < 70. *p*-values < 0.05 were considered statistically significant.

## Data Availability

The study protocol and the individual participant data that underlie the results reported in this article, after de-identification, are available upon request from the corresponding author 6 months after publication. Researchers will need to state the aims of any analyses and provide a methodologically sound proposal. Proposals should be directed to nadja.haiden@meduniwien.ac.at. Data requestors will need to sign a data access agreement and in keeping with patient consent for secondary use, obtain ethical approval for any new analyses.

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
