# Peer review of "Preterm Infants on Early Solid Foods and Neurodevelopmental Outcome—A Secondary Outcome Analysis of a Randomized Controlled Trial"

_nutrients, 2024, doi:10.3390/nu16101528_

Round 1
Reviewer 1 Report
Comments and Suggestions for Authors
Thanhaeuser and colleagues performed a secondary analysis using the dataset obtained from a randomized controlled trial. The difference in the long-term neurodevelopmental outcomes was assessed between two groups of very low birth weight infants, who started solid foods at early and late periods. The authors found no significant difference in the outcomes between the two groups. The study was well designed and conducted at a high standard with the minimum follow-up loss. Potential biases from the follow-up loss were carefully assessed. I have only a few minor points.
Minor points:
It is rather difficult to understand why the outcome can be different according to the starting timing of solid foods. However, this is acceptable considering the nature of secondary analyses of this type.
The backgrounds section should introduce the main findings from the references 3,5 and 10, which significantly help readers understand the rationale of this secondary analysis.
Publisher instruction sentences are left over at the beginning of the Results section.
Author Response
Thanhaeuser and colleagues performed a secondary analysis using the dataset obtained from a randomized controlled trial. The difference in the long-term neurodevelopmental outcomes was assessed between two groups of very low birth weight infants, who started solid foods at early and late periods. The authors found no significant difference in the outcomes between the two groups. The study was well designed and conducted at a high standard with the minimum follow-up loss. Potential biases from the follow-up loss were carefully assessed. I have only a few minor points.
Minor points:
- It is rather difficult to understand why the outcome can be different according to the starting timing of solid foods. However, this is acceptable considering the nature of secondary analyses of this type.
Answer Comment 1: We completely agree with the respected reviewer. We did not expect any differences in neurodevelopmental outcome but included it as a secondary outcome for safety reasons.
- The backgrounds section should introduce the main findings from the references 3,5 and 10, which significantly help readers understand the rationale of this secondary analysis.
Answer Comment 2: We thank the reviewer for the comment. The following sentence is included in the background section.
„The primary outcome of height at one year corrected age did not differ between the study groups [3], and no notable differences were found in secondary outcomes and safety parameters such as iron and vitamin D status during the infants' first year of life [4,5].“
- Publisher instruction sentences are left over at the beginning of the Results section.
Answer Comment 3: Thank you very much for pointing that out, the sentence has been removed.
Reviewer 2 Report
Comments and Suggestions for Authors
I think this study raises an important issue and could be worth further consideration with thorough revision.
1. The output of the LMM is not shown in the manuscript. THis may be provided at least as a supplemental material. I would like to see the fixed and the random effects (random intercepts) and I would like the authors to discuss their meaning. However the LMM is suitable when repeated measurements are available and in the methods this issue is not clear. How many measurements were available per patients on average (provide median and mean number of observation per patient please). Were there missing measurementes? (quantify please) As the authots know, in absence of repeated measurements the LMM becomes equivalent to a general linear model and the random effect part drops. Please the authors clarify the model providing principal coefficients besides the p values.
2. The intrauterine heterogenetity of neurodevelopment was not discussed enough and the reference list needs improvement on this end. Primarily neurodevelopment at two years is massively different depending on the phenotype of PTB. Secondly, there is evidence that neurodevelopment is already altered in utero before birth and is not a mere consequence of preterm birth but anticipates birth being due to some extent to intrauterine environment. Please consider this two important concepts ion the discussion.
Comments on the Quality of English LanguageAdequate
Author Response
I think this study raises an important issue and could be worth further consideration with thorough revision.
1. The output of the LMM is not shown in the manuscript. This may be provided at least as a supplemental material. I would like to see the fixed and the random effects (random intercepts) and I would like the authors to discuss their meaning. However, the LMM is suitable when repeated measurements are available and in the methods this issue is not clear. How many measurements were available per patients on average (provide median and mean number of observations per patient please). Were there missing measurements? (quantify please) As the authors know, in absence of repeated measurements the LMM becomes equivalent to a general linear model and the random effect part drops. Please the authors clarify the model providing principal coefficients besides the p values.
Answer Comment 1: We agree with the respected referee that LMMs are often used for repeated measurements. However, in this study we did not analyze repeated measurements per patient. As stated on page 3 line 130-131 of our manuscript (methods section) we used LMMs with a random intercept to adjust for possible correlation between siblings of multiple births. This is of great importance in a study of preterm infants with a large number of included patients being related as multiple birth twins, triplets, etc., and thereby sharing a common genetic background, more similar than "common" inheritance. Thereby, adjustment for possible correlation between siblings of multiple birth is crucial in this study. This has already been addressed in multiple studies of preterm infants (i.e., Bangma JT, et al. Early life antecedents of positive child health among 10-year-old children born extremely preterm. Pediatr Res. 2019 Dec;86(6):758-765. doi: 10.1038/s41390-019-0404-x.) and methodological papers (i.e., Zou B, et al. A mixed-effects two-part model for twin-data and an application on identifying important factors associated with extremely preterm children's health disorders. PLoS One. 2022 Jun 13;17(6):e0269630. doi:10.1371/journal.pone.0269630).
Furthermore, we used study group, gestational age at birth, sex, nutrition at discharge, highest education of parents, and high grade IVH (defined as > grade II) as fixed effects for our LMMs. All of these were included for empirical reasons. To describe our reasoning for inclusion of these covariates we added the following paragraph to our manuscript (methods section, statistics paragraph):
"Study group (i.e., early vs late complementary feeding) was included to identify differences of neurological outcomes between the randomized studied complementary feeding groups. Gestational age at birth, sex, highest education of parents, and high grad IVH were added as covariates to adjust for potential confounding of these factors on the studied patients neurological outcomes. All of these adjustment factors are on its own known potential influencers of neurological development. This is also true for formula vs mothers own milk diet wherefore the covariate "nutrition at discharge" was added to the model as another potential empirical confounder."
We further added the model coefficients as supplemental material (including fixed and random effects) as well as number of observations per model to address the referees question for quantification of missing measurements. A corresponding note was added to the results section.
“Table 3, supplemental material shows the model coefficents as well as number of observations per model.”
2. The intrauterine heterogeneity of neurodevelopment was not discussed enough, and the reference list needs improvement on this end. Primarily neurodevelopment at two years is massively different depending on the phenotype of PTB. Secondly, there is evidence that neurodevelopment is already altered in utero before birth and is not a mere consequence of preterm birth but anticipates birth being due to some extent to intrauterine environment. Please consider this two important concepts on the discussion.
Answer Comment 2: Thank you for this important comment. We included a paragraph on the topic in the discussion section as suggested by the respected reviewer.
“Furthermore, the intrauterine heterogeneity of neurodevelopment needs to be discussed. Recent studies have focused on the differences in the developmental outcomes of preterm infants dependent on the phenotype of preterm birth [25,26].
Neurodevelopment appears to be altered already in utero due to the intrauterine environment and is therefore more than just a mere consequence of preterm birth. It is becoming increasingly evident that subtle changes in the intrauterine environment can have long-lasting effects on fetal programming and brain development. These changes may be influenced by a number of factors, including maternal nutrition, stress, inflammation, and medications [26-28]. Recent advancements in MRI imaging have un-veiled that the functional connectivity of the fetal brain undergoes modifications prior to preterm birth, with a reduction in fetal brain functional connections observed in infants at a heightened risk of premature delivery [29,30].
However, the influence of prenatal factors on the neurodevelopmental outcomes of preterm infants has not been factored into our analyses. This consideration is pivotal for a comprehensive understanding of the complexity surrounding the neurodevelopmental trajectory of preterm neonates.”
Reviewer 3 Report
Comments and Suggestions for Authors
This comparative study used the data of the prospective trial of very low birth weight infants and investigated whether age (10-12th week vs. 16-18th week) at introduction of solid foods affects neurodevelopmental outcomes in the first three years of life. This study did not find significant differences in neurodevelopmental outcome between groups.
This study provided knowledge for the role of age at introduction of solid foods affects neurodevelopmental outcomes in very low birth weight infants.
The authors may consider revise their manuscript based on the suggestions below.
1. Several abbreviations warrant explanation. For example, “IVH” in Abstract; “ESPGHAN” in Introduction. The abbreviation “PUFA” appeared in line 59 and line 63 repeatedly.
2. The current manuscript did not describe the content of complementary feeding.
3. Did “complementary feeding” and “solid foods” indicated the same things? Was there any different concept between them?
4. I am not sure why the authors emphasized the results of the study on PUFA (line 62-65). Did this study include PUFA as one of components of “solid foods”?
5. How did this study evaluate children’s neurodevelopment on Bayley Scales?
Author Response
This comparative study used the data of the prospective trial of very low birth weight infants and investigated whether age (10-12th week vs. 16-18th week) at introduction of solid foods affects neurodevelopmental outcomes in the first three years of life. This study did not find significant differences in neurodevelopmental outcome between groups.
This study provided knowledge for the role of age at introduction of solid foods affects neurodevelopmental outcomes in very low birth weight infants.
The authors may consider revise their manuscript based on the suggestions below.
- Several abbreviations warrant explanation. For example, “IVH” in Abstract; “ESPGHAN” in Introduction. The abbreviation “PUFA” appeared in line 59 and line 63 repeatedly.
Answer Comment 1: We thank the reviewer for pointing that out. The mentioned abbreviations have been adjusted.
- The current manuscript did not describe the content of complementary feeding.
Answer Comment 2: Thank you. Indeed, we did not include a detailed explanation of the age-appropriated standardized complementary feeding in this paper, because it has been described previously (Haiden, N. & Thanhaeuser, M. et al. Randomized Controlled Trial of Two Timepoints for Introduction of Standardized Complementary Food in Preterm Infants. Nutrients 2022, 14, doi:10.3390/nu14030697.). We now included a sentence about the complementary food provided in the methods section for clarification.
„Five different food boxes following an age-based step-up concept containing commercially available ready-to-use baby jar food were available.“
- Did “complementary feeding” and “solid foods” indicated the same things? Was there any different concept between them?
Answer Comment 3: Thank you for your comment. Yes, “complementary feeding” and “solid foods” indicated the same thing, there was no different concept between them.
- I am not sure why the authors emphasized the results of the study on PUFA (line 62-65). Did this study include PUFA as one of components of “solid foods”?
Answer Comment 4: Thanks again for your comment. We agree with the reviewer that it is a bit confusing and therefore removed the paragraph.
- How did this study evaluate children’s neurodevelopment on Bayley Scales?
Answer Comment 5: We used the German version of the third edition of the Bayley Scales of Infant-Toddler development. The tests were conducted and scored by two certified clinical psychologists with extensive experience in test administration. Please also see the corresponding paragraph in the methods section.
„Neurodevelopmental outcome was assessed at one and two years corrected age, and at three years, four months uncorrected age using the Bayley Scales of Infant-Toddler Development, third edition, German version [16,17]. The Bayley-III consists of five subtests: cognition, receptive and expressive communication, and fine and gross motor skills, and is used to measure the neurodevelopment of infants aged 16 days to 42 months. For each subtest, scaling values are calculated which range from 1 to 19 with a mean of 10 and a standard deviation of 3, based on normative data for the toddlers' ages. The scores are converted into composite cognitive, language and motor scores. These have a mean of 100 and a standard deviation (SD) of 15. Composite scores of -1 SD (values between 70 and 85) are defined as mild disability, scores with -2 SD (values <70) as severe disability. The tests were conducted and scored by two certified clinical psychologists with extensive experience in test administration.“
Round 2
Reviewer 2 Report
Comments and Suggestions for Authors
Acceptable
Comments on the Quality of English LanguageAcceptable